# Reciprocity of social influence

Ali Mahmoodi [1,2], Bahador Bahrami [3,4] & Carsten Mehring [1,2]

Humans seek advice, via social interaction, to improve their decisions. While social interaction is often reciprocal, the role of reciprocity in social influence is unknown. Here, we tested the hypothesis that our influence on others affects how much we are influenced by them. Participants first made a visual perceptual estimate and then shared their estimate with an alleged partner. Then, in alternating trials, the participant either revised their decisions or observed how the partner revised theirs. We systematically manipulated the partner's susceptibility to influence from the participant. We show that participants reciprocated influence with their partner by gravitating toward the susceptible (but not insusceptible) partner's opinion. In further experiments, we showed that reciprocity is both a dynamic process and is abolished when people believed that they interacted with a computer. Reciprocal social influence is a signaling medium for human-to-human communication that goes beyond aggregation of evidence for decision improvement.

[1] Bernstein Centre Freiburg, University of Freiburg, Hansastrasse 9a, 79104 Freiburg, Germany. [2] Faculty of Biology, University of Freiburg, Schänzlestraße 1, 79104 Freiburg, Germany. [3] Institute of Cognitive Neuroscience, University College London, 17 Queen Square London, London WC1N 3AR, UK. [4] Faculty of Psychology and Educational Sciences, Ludwig Maximilian University, Leopoldstrasse 13, 80802 Munich, Germany. These authors contributed equally: Bahador Bahrami, Carsten Mehring. Correspondence and requests for materials should be addressed to A.M. (email: ali.mahmoodi1367@gmail.com)

When we are uncertain, we look for a second opinion and those opinions often change our decisions and preferences[1–5]. Moreover, social influence is not restricted to difficult or critical decisions: evaluations of the comments in the news media are affected by previous scores of the content[6] and risk preferences alter after observing other people's choices[7]. Social influence extends to perceptual judgment[8–10] and long-term memory[11]. Social information can help improve decision accuracy[12], outcome value[13], and evaluative judgment[14–16]. On the other hand, social influence can also lead to catastrophic outcomes. Social influence causes information cascades[17] urging individuals to ignore their own accurate information in favor of the cascaded falsehoods. In some cases, groups are less biased when their individuals resist social influence[18]. Social influence can undermine group diversity[19] leading to disastrous phenomena such as market bubble[20], rich-get-richer dynamics[21], and zealotry[22].

Humans tend to reciprocate in social interaction[23,24]. We react to respect with respect and hostility with hostility[23]. Smiling staff get more tips[25]. Violators of trust in Trust Games[26], and free riders (i.e., those who do not contribute) in Public Good Games are punished[27]. Since social influence is, by definition, mediated via social interaction, one may wonder if reciprocity extends to social influence itself. However, despite a wealth of research on reciprocal behavior, to our knowledge, no study has examined the existence of reciprocity in how we receive and inflict influence on others[28,29].

Several studies in social decision-making in humans have shown a sensible correspondence between the reliability of social information (e.g., advice) and the extent to which that advice is assimilated in decisions and preferences. For example, human agents integrate their own choice with that of an advisor by optimally tracking the trustworthiness of the advisor[30] and are able to track the expertise of several agents concurrently[31]. Social information is integrated into value and confidence judgment based on its reliability[32] or credibility[33]. People can integrate information from themselves and others based on their confidence[9]. These studies offer strong evidence for what previous works in social psychology have called "informational conformity"[34], which is based on the assumption that others can have access to information that will help the agent achieve better accuracy. The prediction drawn from this informational account is that social influence should hinge on the reliability of the source and quality of the advice but not on conventions and norms such as reciprocity. If we could benefit from others' (critical) opinion and the accuracy of our opinion is our only concern, then we should welcome reliable advice irrespective of the advisor's attitude toward our opinion.

On the other hand, numerous studies indicate that people perform less than ideally when using social information[10,35–37]. When aggregating individual and social information about a perceptual decision, humans follow a simplifying heuristic, dubbed equality bias: the tendency to allow everyone equal say in a collective decision irrespective of their differential accuracy or expertise[36,37]. The universal prevalence of the equality bias is important because it shows that even though humans do have the cognitive and computational capacity to track the trustworthiness[30], reliability[31], and credibility[33] of others, they still choose to employ a simple heuristic. Other studies have shown that in a different set of experimental conditions, people show an egocentric bias by relying on their own individual information more than they should[38,39]. These findings suggest that normative concerns such as equality and maintaining a good self-image may also play an important role in interactive decision-making. These factors are not consistent with the Bayesian theory of social information aggregation[30,32,33],

which requires that the influence that people take from others should not depend on norms and conventions. An empirical observation of reciprocity in advice-taking would be inconsistent with the Bayesian theory of social influence[30,32,33].

To summarize, aligning with other people's choices by taking their advice could be motivated informationally[40] to increase accuracy, or normatively[41] to affiliate with others and maintain positive self-esteem. Following other people's advice often leads to more accurate decisions[9,42,43]. Alignment can contribute to a positive self-image[44] and is used as a compensatory tool among minorities[45]. Being ignored in a virtual game can damage one's self-image[46]. Social exclusion has negative consequences on the excluded[47]. We hypothesized that people would reciprocate influence with others because reciprocity is a pervasive social norm[48]. If one breaks the norm of reciprocity, one should expect to be punished[49], for example, by being ignored. Therefore, participants would reciprocate with a reciprocating partner in order to maintain influence over them and avoid the negative experience of losing influence[47]. On the other hand, for a non-reciprocating partner who violates the norm, participants may have a desire to punish the partner by ignoring their opinion. We tested these hypotheses by investigating if participants in a social decision-making task take less advice from insusceptible partners, i.e., those who do not take advice from the participant. Conversely, we also tested if participants take more advice from susceptible partners, i.e., those who are influenced by the participant's suggestions.

We adopted and modified an experimental perceptual task inspired by recent work on aggregation of social and individual information[10]. Participants estimated the location of a visual target on a computer screen. Then they saw the estimate of their partner about the location of the same target. The participant did not know that this partner's opinion was, in reality, generated by sampling randomly from a distribution centered on the correct answer. After the two initial estimates were disclosed, the participant or the partner was allowed to revise their estimate. An algorithm generated the partner's revised estimate simulating susceptible or insusceptible partners. Experiment 1 showed that, participants took more advice from the partner who took more advice from the participant. Experiment 2A and 2B investigated the dynamics of reciprocity and whether reciprocity depends critically on whether we believe it changes the partner's state of mind. Participants thought they worked with either a human or a computer partner and reciprocity disappeared when subjects believed they were working with a computer partner. Finally, our results also showed that reciprocity had a profound impact on participants' evaluation of their own performance, which was lower when working with a non-reciprocating partner.

## Results

**Experiment 1.** In experiment 1, 20 participants were recruited one at a time and told that they would cooperate with three partners who were participating in the same experiment simultaneously in other laboratory rooms connected via internet. In reality, each participant was coupled with a computer algorithm. The algorithm generated three distinct behavioral profiles, corresponding to three experimental conditions (see below), which were administered in a block design in counterbalanced order. Participants were not informed about this arrangement. In each trial, the participant made a perceptual estimate about the location of a target on the screen (Fig. 1). After stating her initial estimate, the participant saw the opinion of the partner about the same stimulus. Next, the participant either revised her estimate or observed the partners revise theirs. Participants were required to put their second estimate between their own first estimate and

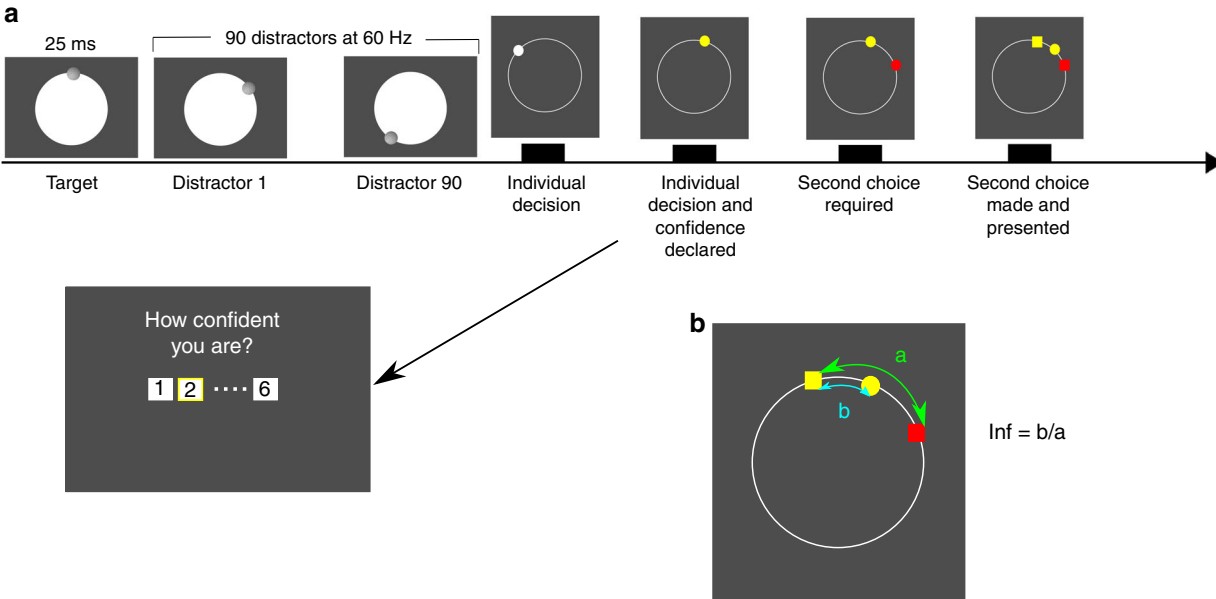

**Fig. 1** Experimental task. **a** Participants first observed a series of dots on the screen. Participants were required to indicate where they saw the very first dot (yellow dot) and then declare their numerical confidence. After making their individual estimates, they were presented with the estimate of a partner (red dot) concerning the same stimulus. After observing the partner's choice, in some trials the participants and in other trials the partner was given a second chance to revise their initial estimate. Afterwards, they were briefly presented with their initial choices and the second choice. In experiment 1, they did the task with three different alleged human partners, which only varied in the second choice strategy in different blocks: in the baseline blocks, the participant made all second choices. In the susceptible blocks, the partner was very influenced by the participant's first choice, however in the insusceptible blocks, the partner was much less influenced by the participant's first choice compared to susceptible blocks. **b** Influence was computed as the angular displacement toward the peer's choice divided by their initial distance from each other's choice

that of their partner's. The acceptable range included staying on their first estimate or moving all the way to their partner's first estimate. Using this constraint, we assured that the amount of change made in the second stage is solely due to observing the partner's choice and not because of a change of mind[50]. The three partners differed in their susceptibility to taking influence from the participant. In the baseline condition, the participant always made the second estimate and the partner never contributed a second estimate. Hence, participants were not able to observe the susceptibility of the baseline partner. In the susceptible condition, the partner was influenced strongly by the participant and revised her initial estimate by conspicuously gravitating toward the participant's estimate. Vice versa, in the insusceptible condition, the partner more or less ignored the participant's opinion. The partner's initial estimate was generated identically in all three conditions by sampling randomly from a distribution centered on the correct answer.

We computed the influence that participants took from their partner as the ratio of the angular displacement (in radians) between their initial and final estimate toward their partner's estimate divided by their initial angular distance from their partner (Fig. 1b). Overall, participants were influenced by their partners' opinions (mean influence ± std. dev. = 0.36 ± 0.11; Wilcoxon sign rank test vs zero, $Z = 3.62$, $p = 0.0002$). Consistent with previous studies[30,33,36], this indicates that our participants did use social information (the partners' choices) to improve their decisions (mean ± std. dev. error after first estimate $67 ± 7$ radians and after second estimate $64 ± 6$ radians, Wilcoxon sign rank test, $Z = 3$, $p = 0.002$). To test our main hypothesis, we asked whether revised opinions were more influenced by the susceptible than the insusceptible (Fig. 2a) and baseline (Fig. 2b) partners. The difference between the average influence in the susceptible and the insusceptible condition, which is a measure of reciprocity, was significantly larger than zero (Fig. 2a–c, Wilcoxon sign rank

test, $Z = 3.33$, $p = 0.002$ after Bonferroni correction). Similarly, the influence from the susceptible partner was larger than the influence in the baseline condition (Fig. 2b, c, Wilcoxon sign rank test, $Z = 2.34$, $p = 0.03$ after Bonferroni correction). The difference between insusceptible and baseline was not significant (Wilcoxon sign rank test, $Z = 1.11$, $p = 0.26$).

The three different partners' initial estimates were produced from an identical generative process using exactly the same distribution and therefore ensuring that the partners' accuracies were perfectly controlled across conditions. However, one might argue that the observed result may be due to the difference in perceived accuracy of the partner. Indeed, we may think more highly of those who confirm our decisions more often and, subsequently, take our (misguided) assessment of their competence as grounds for integrating their estimate into our own revised opinion. To test this hypothesis directly, at the end of each experiment, we asked the participants to rate the precision of the partners they interacted within the experiment, how much they liked different partners, and their own performance as well. A mixed effect model showed that perceived precision of the partners, the actual precision of the partners, the participants' actual precision in different conditions, and the liking of the partner did not have any effect on reciprocity (Supplementary Note 1). The performance ratings of the partner did not differ across conditions (Fig. 2d, repeated measures analysis of variance (ANOVA), $F(2.49, 39) = 1.07$, $p = 0.36$). In each trial, after participants registered their estimates, they were required to report their confidence about their estimates using a scale from 1 to 6. Employing mixed effect models, we showed that condition (baseline, susceptible, or insusceptible) had a significant effect on influence even if confidence was included as a potential confound (Supplementary Note 1).

At the end of the experiment, we asked participants to rate how much they liked their three partners on a scale of 1–10. The mean

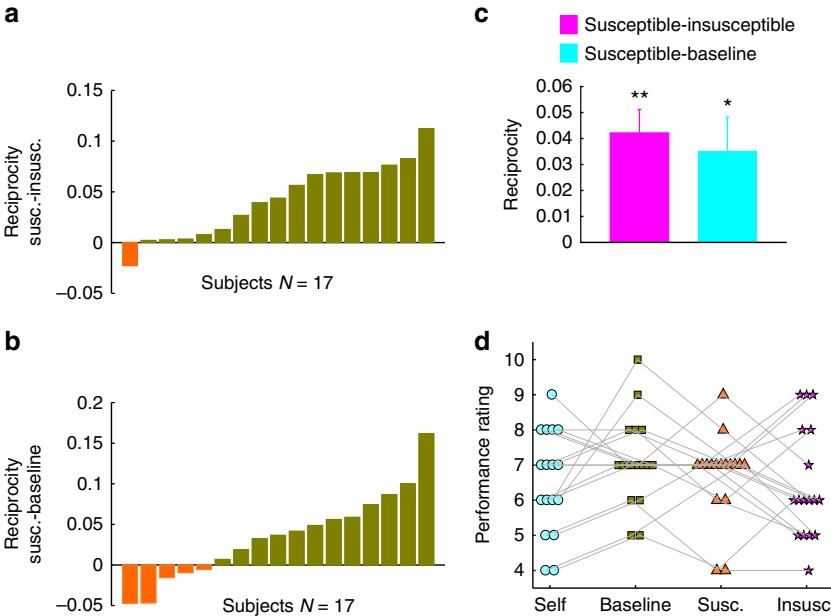

**Fig. 2** Results of experiment 1. **a** Reciprocity, computed as influence in the susceptible condition minus influence in the insusceptible condition, plotted across participants. **b** Reciprocity, computed as influence in the susceptible condition minus influence in the baseline condition, is plotted across participants. **c** Average reciprocity across participants in insusceptible and baseline conditions compared to the susceptible condition. Error bars indicate the standard errors and were computed across the participants' average reciprocities. *$p < 0.05$, **$p < 0.005$; Wilcoxon sign rank test. **d** Performance ratings for self and all partners as reported at the end of the experiment

score ± std. dev. was 7.64 ± 0.7 for the susceptible partner, 5.94 ± 2.53 for the insusceptible partner, and 7.52 ± 1.41 for the baseline partner. Our data shows that people liked the susceptible partner over the insusceptible one (Wilcoxon sign rank test, $Z = 2.62$, $p = 0.008$). There was no difference between susceptible and baseline partners (Wilcoxon sign rank test, $Z = -0.34$, $p = 0.72$) and the $p$-value for the difference between the baseline and the insusceptible partner was only around the threshold (Wilcoxon sign rank test, $Z = -1.91$, $p = 0.056$).

**Experiment 2A and 2B**. The results of our first experiment show that human participants were more influenced by partners which were reciprocally more influenced by the participants. We next asked whether human participants change their advice-taking strategy in response to a change in the advice-taking strategy of a partner over time. To answer this question, we carried out experiment 2A using the same paradigm as in experiment 1 but with an important modification. The participants were told that they are working with the same partner during the entire experiment. The partner's strategy changed across time: in one part of the experiment, the partner was susceptible, in the other part, she was not. Between the two conditions of the experiment, there was a smooth transition (from susceptible to insusceptible or vice versa) and the order of the conditions was counterbalanced across participants (Supplementary Figure 1).

We calculated participants' trial-by-trial influence in the two conditions. Replicating experiment 1, reciprocity, again defined as influence in the susceptible condition minus influence in the insusceptible condition, was significantly larger than zero (Fig. 3a, Wilcoxon sign rank test, $Z = 3.54$, $p = 0.0003$). A mixed effect model showed that condition (susceptible or insusceptible) had a significant effect on influence even if confidence was included as a potential confound (Supplementary Note 1). Again, using a mixed effect model, we showed that the change of influence across conditions cannot be explained by a change in perceived performance of self or partner (Supplementary Note 1).

An interesting question is whether reciprocity is affected by the condition (susceptible or insusceptible) with which the participant started the experiment. To answer this question, we compared the reciprocity for participants who were first exposed to the susceptible partner to participants who started with the insusceptible partner. Our result showed no difference in reciprocity between these two groups (mean ± std. dev. for those who started with reciprocal partner 0.049 ± 0.1 and for those who started with non-reciprocal partner 0.08 ± 0.07, Wilcoxon rank-sum test, $z = -1.07$, $p = 0.28$).

The present results demonstrate that if a participant observes any change in the amount of influence she has over her partner, she will in return modify the amount of advice she takes from her partner. We, therefore, hypothesized that our participants exploit reciprocity as a social signal to communicate with their partner. We predicted that participants would not show reciprocity when working with a computer. In other words, reciprocity depends critically on whether we believe it changes the partner's state of mind. To test this prediction, we conducted experiment 2B in which participants were told that they were working with a computer. All other aspects of the experiment remained as in experiment 2A. As predicted, we did not observe reciprocity when participants believed they were working with a computer (Fig. 3b–d, Wilcoxon sign rank test, $Z = -1.57$, $p = 0.11$). In fact, a majority of participants showed the opposite pattern observed in experiment 2A (Fig. 3b). A two-way ANOVA with factors condition (susceptible or insusceptible as within subjects factor) and type of partner (believed to be human or computer as between subjects factor) and influence as the dependent variable showed a significant interaction of type of partner and condition ($F(1, 58) = 14.8$, $p = 0.00001$). The effect of condition alone was not significant ($F(1, 58) = 0.49$, $p = 0.51$), but there was also a significant between-subject effect of type of partner (Fig. 3c, $F(1, 58) = 4.16$, $p = 0.04$). A post hoc analysis between reciprocity in human and computer condition confirmed a significant difference in reciprocity between these two conditions (Fig. 3c, d, Wilcoxon rank-sum test, $Z = 2.97$, $p = 0.003$).

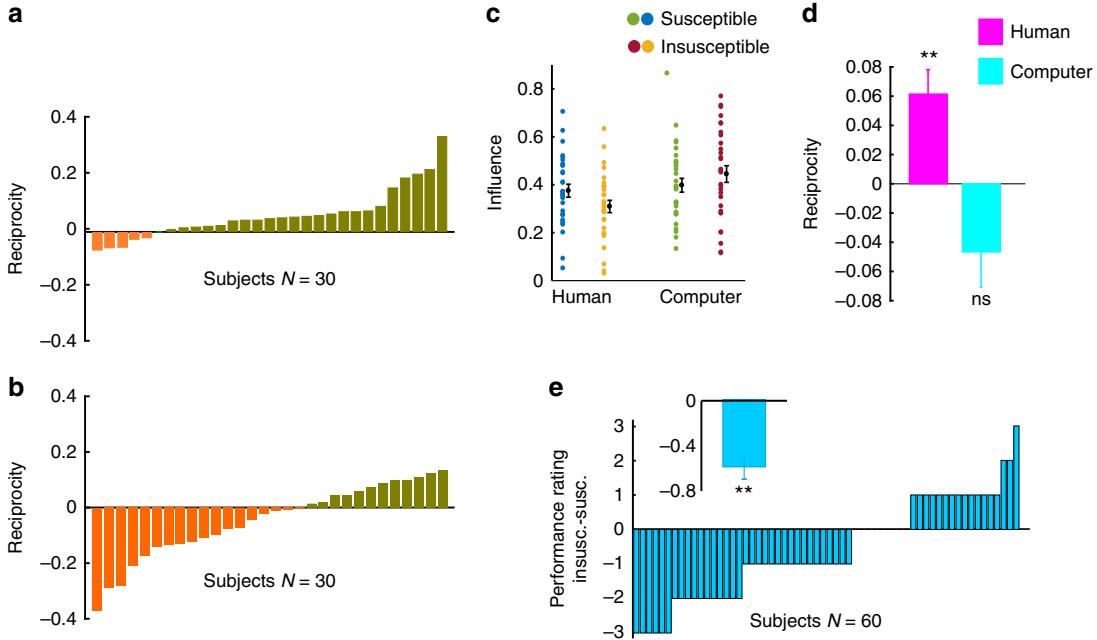

**Fig. 3** Results of experiments 2A and 2B. **a** Reciprocity when participants believe they interact with a human partner. **b** Reciprocity when participants believe they interact with a computer partner. **c** Influence for alleged human and computer partners across susceptible and insusceptible conditions. Dots indicate each participant. Black dots depict the mean influence across participants while error bars depict the standard error of the mean. **d** Average reciprocity across participants when participants think they interact with a human or a computer partner. Error bars depict the standard errors. **e** Difference in participants' performance rating for self, insusceptible minus susceptible, plotted for each participant. The inset shows the mean and standard error across participants. \*\*$p < 0.005$, ns not significant; Wilcoxon sign rank test

Taken together, these results demonstrate that participants were more influenced when they thought their partner is a computer (mean influence ± std. dev.: 42 ± 0.16) compared to when they thought their partner is a human (mean influence ± std. dev.: 0.34 ± 0.13).

We then investigated whether the distance between the participants' and the partners' initial estimates affected the influence that participants took from their partner or the strength of reciprocity (see Supplementary Note 1 for details). We did not find a significant effect of distance (mixed ANOVA, $F_{(2.6, 151)} = 1.17$, $p = 0.31$) nor significant interactions between distance and condition (mixed ANOVA, $F_{(2.6,151)} = 1.47$, $p = 0.22$) or between distance, condition, and experiment ($F_{(2.6, 151)} = 1$, $p = 0.38$). The interaction between distance and experiment was only around the significance threshold ($F_{(2.6, 151)} = 2.7$, $p = 0.05$). We also did not find a significant effect of distance on reciprocity in experiment 2A (repeated measures ANOVA, $F_{(2.61, 75)} = 1.63$, $p = 0.16$). Taken together, these results show that the distance between the initial estimates did not affect the influence that participants took from their partner nor the strength of reciprocity.

Finally, we asked how being in the susceptible and insusceptible conditions changed the participants' ratings of their own and their partner's performance. To answer this question, participants were asked to rate their own and their partners' performance on a 1–10 scale at the end of each condition, i.e., twice in each experiment 2A and 2B. As there was not any difference in performance rating between experiment 2A and 2B, neither for self nor partner (Supplementary Figure 2), we aggregated the data of the two experiments. Participants' rating of their own performance was significantly lower in the insusceptible than in the susceptible condition (Fig. 3e, Wilcoxon sign rank test, $Z = 3.04$, $p = 0.002$), while participants' rating of their partners' performance remained unaffected by the partners' susceptibility (mean rating ± std. dev.: 6.63 ± 1.58 for susceptible condition, and

6.81 ± 1.59 for insusceptible condition; Wilcoxon sign rank test, $Z = -0.9$, $p = 0.36$).

## Discussion

An important question in human social interaction is how people weigh others' opinion[51]. Bayesian theories recommend that different opinions should be weighted by their reliability in order for the group to benefit from putting the opinions together[52]. Indeed, some empirical evidence has supported this view[9,30,32,33] while others have shown other decision aggregation strategies in human social interaction[10,36].

We developed an experimental paradigm inspired by previous work on social information aggregation[10]. We quantified how people weighted their peer's opinion in the context of a visual perceptual task. We tested if this weighting depends on the weight their peers assigned to the participants' opinion. In experiment 1, participants worked with three different alleged human partners in separate blocks. Our participants were more influenced by the susceptible partner compared to the baseline and insusceptible partners. In experiment 2A, the behavior of a single partner changed dynamically within the same experiment from susceptible to insusceptible or vice versa. We replicated the result of experiment 1 by showing that participants were more influenced in the susceptible condition than in the insusceptible condition. In experiment 2B, we showed that participants did not reciprocate when their peer was a computer even though they took greater influence from the computer's advice.

When combining opinions in an optimal Bayesian way to maximize accuracy, each source of information should be weighted based on its reliability[29]. Consequently, reciprocity of social influence, i.e., weighting others' opinion by the weight they give to our opinion is not consistent with Bayesian reliability-based information aggregation. In our experiment, we systematically controlled the accuracy of the participants' partners

such that they were identical across experimental conditions. Participants rated their own performance lower in the insusceptible condition than in the susceptible condition but did not distinguish between the partners' accuracies. With such judgment of their performance and that of their partners, a hypothetical Bayesian participant would have taken more influence from the insusceptible partner. This is actually what participants did when working with a computer partner. However, when working with a human partner, they followed the opposite strategy and took more influence from the susceptible partner.

Why do people go against an information integration strategy that is more likely to maximize their accuracy? We propose that reciprocity is a pervasive social norm[48], and abiding by norms is sometimes rewarding in itself and could hence become a goal[53]. As a consequence, individuals may be ready to pay a cost (in terms of reduced accuracy) to adhere to these norms[54]. This explanation is supported by the finding that participants did not reciprocate with a computer partner as participants did not expect the computer to comply with the reciprocity norm.

In the susceptible condition, participants may reciprocate with their reciprocating partners in order to keep their influence over them and avoid the pain of being ignored[47]. As such, showing reciprocity toward a susceptible partner may be driven by a form of loss aversion. Why would people be aversive to losing influence? Recent works have suggested that influence over others may be inherently valuable both behaviorally[55] and neurobiologically[56]. There is now compelling evidence that others' agreement with our opinion is a strong driver of human brain's reward network[1,57]. In the insusceptible condition, on the other hand, ignoring the insusceptible partner may be motivated by wishing to punish someone who does not comply with the norm of reciprocity.

Following the norm of reciprocity might also improve people's self-efficacy. It is possible that in the insusceptible condition players perceive the experiment as a status competition. In this view, ignoring the insusceptible partners in response to being ignored by them could serve as a signal from the participant that she/he is not willing to accept an inferior position[58,59]. Several studies in behavioral economics (ultimatum game in particular) have shown that one reason why people reject unfair offers is because they want to send the signal that they will not be easily dominated and thereby refuse to accept an inferior social status compared to their peer[60]. Indeed, multiple studies have shown that people do not like to be in an inferior position where their choices are less selected than others' and they use various strategies to compete with their peers in having more influence[56,61,62]. However, people do not engage in a status competition with a computer that is consistent with the difference in reciprocity between human and computer experiments (Fig. 3d). In addition to the above, ignoring the non-reciprocating partner may also serve to protect the participant's "wounded pride"[63] and maintain their self-esteem. Our finding that participants rate their own performance higher when playing with the susceptible than with the insusceptible partner (Fig. 3e) is consistent with the hypothesis that having influence over others improves self-efficacy. It should be noted, however, that, there is no one-to-one correspondence between the perception of self-efficacy and reciprocity: while the difference in self-efficacy between the susceptible and insusceptible conditions is the same for experiment 2A and 2B (Supplementary Figure 2), the difference in influence is not: in experiment 2A, the influence in the insusceptible condition is less than in the susceptible condition but in experiment 2B, the influence in the susceptible condition is identical to the insusceptible condition (Fig. 3c, d). Hence, reciprocity cannot be entirely explained by changes in self-efficacy.

Reciprocating influence is consistent with cognitive balance theory[64], which posits that humans change their preference to be similar to those they like and dissimilar to those they do not.

In experiment 1, participants liked the susceptible partner more than the insusceptible one and were more influenced by the partner they liked more. However, in our experiment, the participants were not allowed to change their estimate away from their peers (participants were instructed to make their second choice between their and their partner's initial estimates). This restriction makes it difficult to directly address the relationship between cognitive balance theory and reciprocity observed in the current study.

In the insusceptible condition, participants' perceived performance of themselves dropped significantly (Fig. 3e). Previous studies show that humans are good at tracking their accuracy even in the absence of any external feedback[9,65]. It is been argued that people are able to get insight into their accuracy through past experience[66]. However, in social contexts, their judgment could be affected by the environment[67] depending on whether they compete or cooperate with a peer[68]. Our performance rating results confirm the effects of social context on human performance monitoring. This finding shows that being ignored exerts a devastating impact on self-efficacy. One possibility is that being repeatedly ignored in the insusceptible condition may induce a negative emotional impression on the participant that impairs the participant's self-evaluation. Another possibility is that participants may interpret the partner's revised estimate as the correct position of the target. By definition, the insusceptible partner's revised estimates would fall further from those of the participant. The inevitable conclusion for the ignored participants would be that their opinion must have been less precise in the insusceptible blocks. Future studies could investigate each of these potential explanations.

Previously, we showed that when working together in a dyad[36], people tend to operate by an "equality bias" giving equal weight to their own and their partner's decision. Participants fulfilled this goal either by adjusting the weight they assign to each other's opinions[36] or by matching their confidence to the confidence of the other people they worked with[61]. Hence, in both cases, people mutually adapted to each other's behavior when required to make decisions together. Similarly, participants exhibited mutual adaptation of social influence in the present study.

Our results imply that humans do not only consider others' reliability to compute the weight that they assign to others' opinion, but instead they take into account other factors like reciprocity as well. We conclude that reciprocity plays a significant role in human advice-taking and social influence, which violates the optimal account of human information integration. Reciprocity as a social norm helps people to fulfill objectives of social interaction including maintaining a positive self-image.

## Methods

**Overview**. A total of 80 healthy adult participants (39 females, mean age ± std. dev.: 25 ± 2.9) participated in three experiments after having given written informed consent. Each participant participated in only one of the experiments. Participants were students at the University of Freiburg, Germany. The experimental procedures were approved by the ethics committee of the University of Freiburg. All experiments were performed using Psychophysics Toolbox[69] implemented in MATLAB (Mathworks). The data were analyzed using MATLAB and SPSS.

**Experimental task 1**. This experiment was designed to investigate whether participants were more influenced by whom they influenced more. Participants first made a perceptual estimate about the location of a target on the screen. Afterwards, they were presented with the estimate from their partner regarding the same stimulus. This was followed by making a second choice about the location of the target or observing a second choice of their partner.

In more detail, the experiment went on as follows: participants ($N = 20$, 9 females, mean age ± std. dev.: 25 ± 2.8) were presented with a sequence of 91 visual stimuli consisting of small circular Gaussian blobs ($r = 5$ mm) in rapid serial visual presentation on the screen (resolution = 2560 × 1440 Dell U2713HM 27″). The first item was presented for 30 ms and every other stimulus

was presented for 15 ms each. Participants' task was to identify the location of the first stimulus. Participants were required to wait until the presentation of all stimuli were finished, and then indicate the location of the target stimulus using the computer mouse (Fig. 1). The reported location was marked by a yellow dot. After participants reported their initial estimate, they were required to report their confidence about their estimate on a numerical scale from 1 (low confidence) to 6 (high confidence). Afterwards, participants were shown the choice of their partners about the same stimulus (see below, Constructing partners section for further details) by a small dot on the screen. Then, either the participant revised her estimate or observed the partner revise theirs. After the second estimate was made, all estimates were presented to the participant for 3 s. In this stage, the first choice was shown by a hexagon to be distinguished from the second choice, which was shown by a circle (Fig. 1). There was not any time pressure on participants in any stage of the experiment and the experiment did not move to next stage until the participants had registered their responses (Fig. 1). Participants were told that their payoff will be calculated based on their first and second estimates. However, everyone was given a fixed amount at the end of the experiment.

During the course of the experiment, participants were exposed to three different partners. Partners varied in their susceptibility to the participants' estimates (i.e., the amount of influence the participants' first estimate has on the partner's second estimate). In the baseline blocks, the participant always made the second estimate and the partner never contributed a second estimate. In the susceptible and insusceptible blocks, the partner and the participants made the second estimate in the odd and even trials, respectively. In the susceptible block, the partner was influenced strongly by the participant and vice versa in the insusceptible block (see below for details in the section "Constructing partners"). In each block, the participants worked with only one partner and each block contained 30 trials. Participants worked with each partner for five blocks. For example, they worked with baseline partner in block 1, then with the susceptible partner in block 2, then with insusceptible partner in block 3, then again with the baseline partner in block 4, and so on. Participants completed 15 blocks in total. The order of the partners was randomized across participants. The three partners were shown by blue, red, and turquoise markers, which were randomly assigned to the different partners at the beginning of each subject's experiment. After finishing the experiment, the participants were required to estimate their own and the three partners' performance on a numerical scale from 1 to 10. They were instructed to only consider the first choice to assess their partners' performance. At the end, we asked them whether they thought they interacted with real people or with a computer algorithm. All participants indicated that they believed they were interacting with real human partners.

Three participants were excluded from the final analyses of this experiment. One participant did not notice she played against three different partners. The other two participants were excluded because they resampled in the second stage, meaning that their second estimates were not between their own and their partner's initial estimates (contrary to the task instruction). In the Supplementary Figure 3, we show that our findings remain valid and statistically significant when these three subjects were not excluded from the analysis.

**Experimental task 2A**. This experiment was designed to test whether reciprocity is a dynamic process. The experiment used the same paradigm as experiment 1 but participants ($N = 30$, 15 females, mean age ± std. dev.: 25 ± 2.8) were told that they do the task with only one partner, which is the same gender as themselves. The partner changed its susceptibility during the course of the experiment, either from susceptible to insusceptible or vice versa. The experiment consisted of 11 blocks in total. Half of the participants were first probed in the susceptible condition, which lasted five blocks and then with a transition block in between, they switched to five insusceptible blocks. The other half completed the opposite order. The average advice/influence that the partner took from our participants is depicted in Supplementary Figure 1. The transition block was designed in order to avoid a sudden change of the partner's behavior. During the transition block, the partner's advice-taking strategy linearly (see below) switched from susceptible to insusceptible or vice versa.

**Debriefing**. After each session of the experiment, all participants were debriefed to assess to what extent they believed the cover story. We interviewed them with indirect questions about the cover story and all participants stated that they believed they were working with other human participants in neighboring experimental rooms.

**Experimental task 2B**. This experiment differed from experiment 2A in one respect: participants ($N = 30$, 15 females, mean age ± std. dev.: 24 ± 3.1) were told that their partner in the experiment is a computer. Any other aspects of the experiment were identical to experiment 2A and they received exactly the same task instructions as in experiment 2A except that the human partner was replaced by a computer.

**Performance rating**. In experiment 1, participants rated their own and the three different partners' performance once at the end of the experiment. Note that a different color identified each partner during the experiment. In experiment 2A

and 2B, participants rated their own and their partner's performance at the end of each block. This way, we obtained a pair of performance ratings for the self and the susceptible partner and another pair for the self and the insusceptible partner.

**Constructing partners**. The error distribution of all partners' first choices was modeled from participants' actual estimation errors during a pilot experiment. Ten participants performed an experiment identical to experiment 1 of the current study. We aggregated errors of all participants ($N = 10$) and fitted the concentration parameter kappa of a von Mises distribution centered to the target, yielding the value kappa = 7.4. Then, in each trial, we drew the first choice of the partner from this distribution. We speculated that participants' assessment of their partners' performance may be strongly influenced by the few trials with high confidence (confidence level of 5 or 6). To avoid this potential problem, the partner's first choice was not taken from the von Mises distribution in high confidence trials but randomly drawn from a uniform distribution centered on the participants' choice with a width of ±20°.

The second choice of the partner was computed differently for susceptible and insusceptible partners. For experiment 1, the influence that the insusceptible partner took from the participants in each trial was chosen with a probability of 0.65 from a uniform distribution on the interval [0, 0.2], with a probability of 0.2 randomly from a uniform distribution on the interval [0.3, 0.7], and with a probability of 0.15 randomly from a uniform distribution on the interval [0.7, 0.9]. For experiment 2, the influence that the insusceptible partner took from the participants was chosen randomly from a uniform distribution on the interval [0, 0.2]. For the susceptible partner, in all experiments, the influence was chosen with a probability of 0.5 randomly from a uniform distribution on the interval [0.7, 1], with a probability of 0.2 randomly from a uniform distribution on the interval [0.3, 0.7], and with a probability of 0.3 randomly from a uniform distribution on the interval [0, 0.3]. In the transition block, the influence of the partner was a linear interpolation between the susceptible and the insusceptible partner:

$$\text{inf} = (1 - \lambda) \times \text{inf}_\text{s} + \lambda \times \text{inf}_\text{ins},$$

where $\text{inf}_\text{s}$ and $\text{inf}_\text{ins}$ were the influences of the susceptible and insusceptible partners, respectively (as explained above). $\lambda$ gradually increased with time from 0 at the beginning to 1 at the end of the transition block for a transition from the susceptible to the insusceptible condition. For the transition from the insusceptible to the susceptible condition, $\lambda$ decreased gradually from 1 to 0.

In experiment 1, on average, the advice that the partners took from the participants was 0.3 and 0.55 in the insusceptible and susceptible conditions, respectively. In experiment 2A, on average, the advice that the partner took from the participants was 0.07 and 0.5 in the insusceptible and susceptible conditions, respectively. The second choice of the partners in experiment 2B was designed exactly the same as in experiment 2A and the average advice that the partner took in the insusceptible and the susceptible condition was identical to experiment 2A.

**Computation of error bars**. All error bars in the figures depict standard errors. Standard errors were computed across subjects; in situations where we had multiple measurements from each participant, we first computed the mean value for each participant, and then computed the standard error across the participants' mean.

**Data availability**. The behavioral data that support the findings of this study and the code that was used to generate the findings and to conduct the experiments of this study will be provided to all readers upon request.

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

## Acknowledgements

This work was supported by a PhD scholarship (A.M.) from the Graduate School Scholarship Program of the German Academic Exchange Service (DAAD), a European Research Council Starting Grant "NeuroCoDec #309865" (B.B.), the German Research Foundation (DFG, grant no INST 39/1014-1 FUGG) (C.M.) and the "Struktur -und Innovationsfonds Baden-Württemberg (SI-BW)" of the state of Baden-Württemberg (C.M.). We thank Ulf Toelch for helping to implement the experimental task and Helena Gavrilova, Tobias Pistohl, and Luke Bashford for helping to collect data.

## Author contributions

A.M., B.B., and C.M. designed the experiments. A.M. collected the data. A.M. carried out the data analysis. A.M., B.B., and C.M. interpreted the results and wrote the manuscript.

## Additional information

**Competing interests:** The authors declare no competing interests.

