## [Peer Review File · Nature Communications]

Reviewers' comments:

Reviewer #1 (Remarks to the Author):

The authors explore an interesting topic - to what extent are subjects influenced by others depending on whether these others have shown to be susceptible or unsusceptible to be influenced by the subject? The authors find that subjects are influenced by susceptible partners but not unsusceptible ones. They also find that subjects, when being told that the partner is either a computer or a human, are not influenced by computers.

While I find the overall topic very interesting, I have some problems with the paper.

On the updating: Some types of updating/reacting to the other person's choice would just not make sense in this setting - if I have a clear idea in what ball park the target was, and I see something completely different from my partner, I might just not update no matter how susceptible my partner is. I would like to see more analysis on whether people are less influenced the further the partner's estimate was no matter what, and how this also interacts with treatment. In a related vein, I also find the whole framing of the paper to be problematic. The reciprocity/cooperation literature is almost irrelevant here (in the introduction and discussion). Instead, the literatures on Bayesian updating (or the lack of it) as well as imitation or conformity are very relevant and I would like to know more about the contribution of this specific paper to those literatures. I also don't understand what the authors did in terms of incentivizing participants to care about the task. There ultimately were no incentives here for subjects to get their estimates right, so then I'm not sure what we really learn here when we see that people sometimes (at no cost) react to how much their partner reacts to their estimates. Why did the authors choose this setup?

On the analysis: how are standard errors treated? By clustering them at the subject level or through some fixed effects model? This matters for the interpretation of p-values.

Other comments:

There's basically nothing in the introduction about social influence also having the possibility to lead to bad outcomes. This should be updated.

Did the subjects believe that they actually were linked to other real participants? Or is this a lab where the subjects are used to being lied to? That should affect behavior and interpretations of results.

I don't understand why the authors chose to exclude three participants from the analysis from experiment 1. Why does it matter if the participant does not notice she played against three different partners? Maybe she only cared about their play and not who they were. Or if participants resampled in the second stage - I don't see that as a reason to exclude any observations. Maybe they distrusted their own estimate and that of their partner.

Given the known problems with p-value thresholds like 0.05, I would skip discussing $p < 0.10$ as "marginally significant" and $p < 0.15$ as "a non-significant trend".

Reviewer #2 (Remarks to the Author):

This is a well written manuscript that tests a simple but intriguing idea - that we take advice from people who also take advice from us. Mahmoodi and colleagues use a laboratory task to carefully

measure very straightforward perceptual judgements made by people who believe they are performing the task in the company of another person. Despite the simplicity of perceptual judgements, they are systematically influenced by the perceptual reports of the other agent. However, this happens to a greater degree when the other person is, themselves, influenced by the subject him- or herself. The key result is summarized in figure 2. Figure 3 shows that the result is not found when the subject believes they are playing with a computer rather than another person even though exactly the same series of events occur in both conditions.

1. I wondered if it might be appropriate to cite a recent paper (Bang et al., Nat Hum Behavior, 2017) and discuss the relationship between the current study and the previous one a little more. This seems especially appropriate as some of the current authors are authors of this previous study. Bang and colleagues use the Bahrami 2012 paradigm (deciding between one of two locations with Gabor patches) where two players indicate their choice and confidence. Bang and colleagues find that participants match their mean level of confidence. However, importantly, the task is designed so that the decision with the higher indicated confidence is automatically picked as the group decision. So matching confidence leads to giving both partners an overall equal influence on the group decision. Overall both players will have decided in 50% of the cases for the group. These findings therefore also seem to indicate that people adapt to each other during group decision making and reciprocate in terms of their influence on the other one and the decision of the whole group.

2. One minor issue: It seems that subjects were instructed to always make their second judgement towards the other player. Page 11, 304-306:

The other two participants were excluded because they resampled in the second stage, meaning that their second estimates were not between their own and their partner's initial estimates (contrary to the task instruction).

The main finding seems still valid, but the fact that subjects were instructed in that way makes it slightly less surprising. I think that it is important that this feature of the procedure is not just mentioned in the details of the Methods but that it is highlighted at the point in the Results where the main behavioural result is first mentioned (maybe somewhere near page 4?).

If the subjects' decisions had any other implications for their partner then this should be mentioned too.

3. Figure 3c – is the case that the overall influence of the computer was greater than a human partner (even if the reciprocity effect was bigger with the human partners)? Maybe a brief mention of this feature of the results would be useful.

Minor points

Line 50: "a positive self-esteem" should be "positive self-esteem"

Line 52 "is devised as a compensatory tool among minorities" should perhaps be "is used as a compensatory tool by minorities"

Line 58 "dissimilar to those we disapprove" should be "dissimilar to those we disapprove of"

Line 64 "by recent works" should be "by recent work"

Line 103 "by comparing influence under susceptible to the baseline condition" is unclear

Line 388 "Reciprocity when participants believe to interact with a human partner" should be "Reciprocity when participants believe they interact with a human partner"

Reviewer #3 (Remarks to the Author):

Mahmoodi et al. present here a compelling study. There is much I like about this work. I find the question interesting, the experiments simple and elegant, and the methods and analysis appropriate and robust (replication, correction for multiple comparisons, testing interaction effects directly using all data etc.). Moreover, the manuscript is concise and well written.

Please find my questions and comments below:

1- The authors suggest cognitive balance theory as a potential explanation of the findings. If this is the case, the findings can be reduced, in my mind, to two simple steps: 1. I like people that don't ignore me, more than those who do. 2. I change my behavior to align with the actions of someone I like, and vice versa (à la cognitive balance).

However, looking at the results I am not sure this interpretation holds. Specifically, when compared to baseline, the results show that participants change their behavior only towards people that reciprocate (potentially because the subjects cannot move away from the other's rating) but liking ratings are different only for people that don't reciprocate. Even if this doesn't hold, I don't see this necessarily as a weakness since each of the aforementioned steps has been demonstrated multiple times in the past and verifying them under perhaps a different context is not, in my opinion, the strength of the current work.

2- I like experiment 2 that goes beyond general effects to provide a temporal manipulation. As before the design is elegant and thought-out (for example including the transition block, using a different distribution for high confidence answers etc.). I was not sure however why rating data were aggregated across both the computer and human conditions which show opposite behavioral effects. How can one interpret the meaning of these rating differences in this case?

Indeed the result that when your opinion is ignored, your perception of self-efficiency is lower makes sense, but this seems to be associated with less behavioral change in experiment 2a and more change in experiment 2b. Thus, I don't see a simple relationship between the perception of self-efficiency and reciprocity, and I am not sure joining these groups makes sense conceptually. It could also be helpful to see the baseline condition here as well, for comparison.

Additionally, the actual elicitation of these ratings was not clear to me –from the methods and fig. 2D it seems that subjects only rated their own performance once at the end of the experiment, but in the final paragraph of the results this is broken down to at least 2 ratings (self-performance in the different blocks).

Finally, I would be interested to learn if there is some asymmetry depending on whether you start with a reciprocating partner or a non-reciprocating partner. I think this could be an interesting addition for understanding the dynamics and have practical implications.

3-I found some of the text in the introduction and discussion confusing. For example, I am not sure I fully grasp the distinction between the two reasons proposed in the introduction for reciprocation ("a pervasive social norm" and "improve our self-image... help avoid being excluded"). Isn't part of the reason individuals follow pervasive norms exactly to improve their self-image and avoid exclusion?

More generally I felt the relationship between the interpretations and the actual results was not always clear to me (see also comments above). For example, from the introduction, I understood that one may reciprocate because this will reduce the chance of exclusion and even that one may reciprocate because she knows that this will lead the other side to also reciprocate and thus her self-image will not be harmed. However, I am not sure I see how this line of reasoning causally explains the main effects described here (relating to the differences between the conditions). It may even lead to the opposite prediction since if I want to protect my self-image and avoid exclusion I may reciprocate more to people whose actions indicate I am currently in a high risk of exclusion.

In the discussion, the authors break this down a bit more to suggest different interrelated motivations under the different conditions (e.g. a reactive effect that helps protect the self-image under the insusceptible condition and affiliation under the susceptible condition). I feel the distinction between these motivations in this paradigm as well as the evidence linking these claims to the results is not sufficiently clear currently.

On the other hand, I find the distinction between normative and informational much clearer and very nicely controlled in this task.

4- Does the asymmetry mentioned in comment 1 replicate in experiment 2a?

5- It seems that the influence the insusceptible partner can take is randomly selected between 0-0.3 but that the average taken in experiment 1 is 0.3. Is this a typo?

Reviewers' comments:

Reviewer #1 (Remarks to the Author):

The authors explore an interesting topic - to what extent are subjects influenced by others depending on whether these others have shown to be susceptible or unsusceptible to be influenced by the subject? The authors find that subjects are influenced by susceptible partners but not unsusceptible ones. They also find that subjects, when being told that the partner is either a computer or a human, are not influenced by computers.

While I find the overall topic very interesting, I have some problems with the paper.

We thank the reviewer for his/her careful review and we were happy to hear that this reviewer considers the "overall topic very interesting".

In the revised manuscript, we have added further analyses, rewrote the introduction, discussion, and result sections to address the reviewer's comments. In the following we address each of the reviewer's comments point by point.

On the updating: Some types of updating/reacting to the other person's choice would just not make sense in this setting - if I have a clear idea in what ball park the target was, and I see something completely different from my partner, I might just not update no matter how susceptible my partner is. I would like to see more analysis on whether people are less influenced the further the partner's estimate was no matter what, and how this also interacts with treatment.

We thank the reviewer for this suggestion. To address the reviewer's question we added a new analysis to the revised manuscript in which we investigate the dependence of influence and reciprocity on the distance between the participant's and their partner's initial estimates. We did not find any significant effect of distance on reciprocity or influence, neither in the human nor in the computer experiment (experiments 2A and B). Hence, how the influence changes across experiments is distance independent. We have added these results to the result section of the main text (lines 204-213) and a more detailed description of the analyses and results to the supplementary material (lines 94-112).

In a related vein, I also find the whole framing of the paper to be problematic. The reciprocity/cooperation literature is almost irrelevant here (in the introduction and discussion). Instead, the literatures on Bayesian updating (or the lack of it) as well as imitation or conformity are very relevant and I would like to know more about the contribution of this specific paper to those literatures.

We have substantially revised and extended our introduction and discussion sections. We now discuss studies which have shown that humans are optimal at integrating social and nonsocial information as

well as studies which have challenged this view (lines 52-77). In addition, we discuss the work on conformity which is relevant to our study. Moreover, we have added a paragraph to discuss how our study contributes to our understanding of the relationship between optimality and reciprocity (lines 240-250).

I also don't understand what the authors did in terms of incentivizing participants to care about the task. There ultimately were no incentives here for subjects to get their estimates right, so then I'm not sure what we really learn here when we see that people sometimes (at no cost) react to how much their partner reacts to their estimates. Why did the authors choose this setup?

We would like to clarify that there was an incentive for the subjects to get their estimates right. In the written instruction which were handed out to the subjects prior to the experiment, it was explained that their reward at the end of the experiment would be calculated based on their accuracy in the first and second estimation stage. This was explained in the methods section of the previous submission and can now be found in lines 345-346 in the revised manuscript.

On the analysis: how are standard errors treated? By clustering them at the subject level or through some fixed effects model? This matters for the interpretation of p-values.

We thank the reviewer for drawing our attention to this point. All error bars show standard errors across participants. We have added a corresponding explanation to the figure captions.

Other comments:

There's basically nothing in the introduction about social influence also having the possibility to lead to bad outcomes. This should be updated.

We thank the reviewer for this suggestion. We have now extended our introduction by referring to examples where social influence can lead to bad outcomes including market bubbles, rich get richer dynamics and zealotry. We also included examples where social influence can undermine the wisdom of crowds, by herding or by reducing group diversity (lines 41-45).

Did the subjects believe that they actually were linked to other real participants? Or is this a lab where the subjects are used to being lied to? That should affect behavior and interpretations of results.

After the experiment, we debriefed all participants by indirect questions and they all believed that they had been doing the task with other human participants in other rooms. This is now explained in more

detail in the Methods section (lines 382-385). Around the time our experiments took place there were no other experiments in the lab in which a cover story was used.

I don't understand why the authors chose to exclude three participants from the analysis from experiment 1. Why does it matter if the participant does not notice she played against three different partners? Maybe she only cared about their play and not who they were. Or if participants resampled in the second stage - I don't see that as a reason to exclude any observations. Maybe they distrusted their own estimate and that of their partner.

Most importantly, when these three subjects are included in the analysis, our findings remain valid and statistically significant. We report this additional result in the Supplementary Material of the revised manuscript (lines 132-144) and refer to it in the methods section of the main text (lines 368-370).

In addition, we further elaborate the reasons why we excluded these subjects:

We asked the participants to stay within the area between their own and their partner's first estimate because this would allow us to merely compute participant's malleability to their partner's estimates. However, in situations when they moved away from their partner's and their own estimates, it is very tricky to interpret their second estimate; we can't claim they did so after exposing to their partner's estimate because they got away from that and hence we can't consider this change as influence and we can't measure influence in these trials. That was our main reason to exclude two subjects who showed this kind of behavior.

Regarding the participant who did not notice that she was playing with three different players, in the debriefing, we noticed that she did not pay attention to the task instruction and did not even pay attention to the estimates of her partners and was distracted during the experiment. We therefore decided to exclude this subject.

Given the known problems with p-value thresholds like 0.05, I would skip discussing $p < 0.10$ as "marginally significant" and $p < 0.15$ as "a non-significant trend".

We agree with the reviewer and have removed the corresponding statements from the text (lines 160-161 and line 192).

Reviewer #2 (Remarks to the Author):

This is a well written manuscript that tests a simple but intriguing idea – that we take advice from people who also take advice from us. Mahmoodi and colleagues use a laboratory task to carefully

measure very straightforward perceptual judgements made by people who believe they are performing the task in the company of another person. Despite the simplicity of perceptual judgements, they are systematically influenced by the perceptual reports of the other agent. However, this happens to a greater degree when the other person is, themselves, influenced by the subject him- or herself. The key result is summarized in figure 2. Figure 3 shows that the result is not found when the subject believes they are playing with a computer rather than another person even though exactly the same series of events occur in both conditions.

We thank the reviewer for the positive assessment of our work and for her/his careful review.

1. I wondered if it might be appropriate to cite a recent paper (Bang et al., Nat Hum Behavior, 2017) and discuss the relationship between the current study and the previous one a little more. This seems especially appropriate as some of the current authors are authors of this previous study. Bang and colleagues use the Bahrami 2012 paradigm (deciding between one of two locations with Gabor patches) where two players indicate their choice and confidence. Bang and colleagues find that participants match their mean level of confidence. However, importantly, the task is designed so that the decision with the higher indicated confidence is automatically picked as the group decision. So matching confidence leads to giving both partners an overall equal influence on the group decision. Overall both players will have decided in 50% of the cases for the group. These findings therefore also seem to indicate that people adapt to each other during group decision making and reciprocate in terms of their influence on the other one and the decision of the whole group.

We thank the reviewer for this suggestion. We have updated our discussion and have addressed the reviewer's points in lines 306-311.

2. One minor issue: It seems that subjects were instructed to always make their second judgement towards the other player. Page 11, 304-306: The other two participants were excluded because they resampled in the second stage, meaning that their second estimates were not between their own and their partner's initial estimates (contrary to the task instruction). The main finding seems still valid, but the fact that subjects were instructed in that way makes it slightly less surprising. I think that it is important that this feature of the procedure is not just mentioned in the details of the Methods but that it is highlighted at the point in the Results where the main behavioural result is first mentioned (maybe somewhere near page 4?). If the subjects' decisions had any other implications for their partner then this should be mentioned too.

We agree with the reviewer and we have revised the results according to his/her suggestions (lines 114-118). The subjects were told that in the second stage they can change their first estimate given the first estimate of their partner and were told that everyone else were given the same instruction. Thus the subjects' decision did not have any other implications for their partner.

3. Figure 3c – is the case that the overall influence of the computer was greater than a human partner (even if the reciprocity effect was bigger with the human partners)? Maybe a brief mention of this feature of the results would be useful.

The reviewer is right that the overall influence of the computer partner was higher than for the human partner. This result is described on lines 198-199 in the manuscript.

Minor points

Line 50: “a positive self-esteem” should be “positive self-esteem”

Line 52 “is devised as a compensatory tool among minorities” should perhaps be “is used as compensatory tool by minorities”

Line 58 “dissimilar to those we disapprove” should be “dissimilar to those we disapprove of”

Line 64 “by recent works” should be “by recent work”

Line 103 “by comparing influence under susceptible to the baseline condition” is unclear

Line 388 “Reciprocity when participants believe to interact with a human partner” should be “Reciprocity when participants believe they interact with a human partner”

We thank the reviewer for these corrections and we have modified the manuscript accordingly.

Reviewer #3 (Remarks to the Author): Mahmoodi et al. present here a compelling study. There is much I like about this work. I find the question interesting, the experiments simple and elegant, and the methods and analysis appropriate and robust (replication, correction for multiple comparisons, testing interaction effects directly using all data etc.). Moreover, the manuscript is concise and well written.

We thank the reviewer for positive assessment and careful review of our work.

Please find my questions and comments below:

1- The authors suggest cognitive balance theory as a potential explanation of the findings. If this is the case, the findings can be reduced, in my mind, to two simple steps: 1. I like people that don't ignore me, more than those who do. 2. I change my behavior to align with the actions of someone I like, and vice versa (à la cognitive balance). However, looking at the results I am not sure this interpretation holds.

Specifically, when compared to baseline, the results show that participants change their behavior only towards people that reciprocate (potentially because the subjects cannot move away from the other's rating) but liking ratings are different only for people that don't reciprocate. Even if this doesn't hold, I don't see this necessarily as a weakness since each of the aforementioned steps has been demonstrated multiple times in the past and verifying them under perhaps a different context is not, in my opinion, the strength of the current work.

We agree with the reviewer that the main point of this paper is not the confirmation of cognitive balance theory but instead the presence of reciprocity in human social influence. We therefore, rephrased the section on cognitive balance theory in the discussion and included the reviewer's points (lines 289-292).

2- I like experiment 2 that goes beyond general effects to provide a temporal manipulation. As before the design is elegant and thought-out (for example including the transition block, using a different distribution for high confidence answers etc.). I was not sure however why rating data were aggregated across both the computer and human conditions which show opposite behavioral effects. How can one interpret the meaning of these rating differences in this case?

We thank the reviewer for bringing this issue up. Since the difference in performance rating between susceptible and insusceptible conditions were the same in experiment 2A and 2B, we aggregated the data of the two experiments. However, we now report the rating differences separately for experiments 2A and 2B (incl. separate figures) in the Supplementary Material (lines 114-130). We have also revised the corresponding section in the main text accordingly (lines 217-218).

Indeed the result that when your opinion is ignored, your perception of self-efficiency is lower makes sense, but this seems to be associated with less behavioral change in experiment 2a and more change in experiment 2b. Thus, I don't see a simple relationship between the perception of self-efficiency and reciprocity, and I am not sure joining these groups makes sense conceptually. It could also be helpful to see the baseline condition here as well, for comparison.

We agree with the reviewer that there is no simple relationship between the perception of self-efficacy and reciprocity. This is supported by the fact that while the difference in self-efficacy between the susceptible and insusceptible conditions is the same for experiment 2A and 2B (see Figure S2 in the supplementary material), the difference in influence is not the same in both experiments. Although in experiment 2A, the influence in the insusceptible condition is less than in the susceptible condition, in experiment 2B, the influence in the susceptible condition is identical to the insusceptible condition (Figure 3C, D). Hence, reciprocity cannot be entirely explained by changes in self-efficacy.

In our discussion we, therefore, only discuss that being ignored in the insusceptible condition can be the reason for participants' reduced self-efficacy but we do not suggest that reciprocity can be explained by self-efficacy. We have revised the discussion to clarify these points (lines 277-285).

Please note that in experiment 2A and 2B, we had only susceptible and insusceptible conditions and we did not have a baseline condition.

Additionally, the actual elicitation of these ratings was not clear to me –from the methods and fig. 2D it seems that subjects only rated their own performance once at the end of the experiment, but in the final paragraph of the results this is broken down to at least 2 ratings (self-performance in the different blocks).

In experiment 1, participants rated their own and three different partners' performance once at the end of the experiment. However, in experiment 2A and 2B, participants rated their own and their partner's performance once at the end of susceptible blocks and once at the end of insusceptible blocks. We clarify this in the revised manuscript where we present the result of the performance rating (for experiment 1, lines 145-147 and for experiments 2A and 2B, lines 215-217). We have also added a separate section in the Methods section (lines 390-394) which addresses this issue.

Finally, I would be interested to learn if there is some asymmetry depending on whether you start with a reciprocating partner or a non-reciprocating partner. I think this could be an interesting addition for understanding the dynamics and have practical implications.

We performed an additional analysis and found no asymmetry depending on the partner with which you start the experiment. We added this finding to the revised manuscript (lines 178-184).

3-I found some of the text in the introduction and discussion confusing. For example, I am not sure I fully grasp the distinction between the two reasons proposed in the introduction for reciprocation (“a pervasive social norm” and “improve our self-image... help avoid being excluded”). Isn't part of the reason individuals follow pervasive norms exactly to improve their self-image and avoid exclusion?

We agree with the reviewer that part of the reason people follow norms is to avoid being excluded and improve self-efficacy. We have rephrased the introduction to address this point (lines 83-85).

More generally I felt the relationship between the interpretations and the actual results was not always clear to me (see also comments above). For example, from the introduction, I understood that one may reciprocate because this will reduce the chance of exclusion and even that one may reciprocate because she knows that this will lead the other side to also reciprocate and thus her self-image will not be

harmed. However, I am not sure I see how this line of reasoning causally explains the main effects described here (relating to the differences between the conditions). It may even lead to the opposite prediction since if I want to protect my self-image and avoid exclusion I may reciprocate more to people whose actions indicate I am currently in a high risk of exclusion. In the discussion, the authors break this down a bit more to suggest different interrelated motivations under the different conditions (e.g. a reactive effect that helps protect the self-image under the insusceptible condition and affiliation under the susceptible condition). I feel the distinction between these motivations in this paradigm as well as the evidence linking these claims to the results is not sufficiently clear currently. On the other hand, I find the distinction between normative and informational much clearer and very nicely controlled in this task.

We apologize for this confusion. We have now rephrased the introduction and discussion substantially to clarify our interpretation. We now state that the motivation behind reciprocating may be different in susceptible and insusceptible conditions. In both cases, our hypothesis rests on a common assumption that reciprocity is a widely upheld social norm¹ and if one breaks the norm of reciprocity, one should expect to be punished, for example by being ignored.

In the susceptible condition, participants may reciprocate with their partners in order to *maintain* their influence over them and avoid the *negative experience of losing* influence. As such, showing reciprocity towards a susceptible partner may be driven by a form of loss aversion. Why would people be aversive to losing influence? Recent works have suggested that influence over others may be inherently valuable both behaviorally and neurobiologically². There is now compelling evidence that others' agreement with your opinion is a strong driver of human brain's reward network^{3,4}. In addition to being inherently valuable, we also conjecture that having social influence may also protect one's self-efficacy, a claim which we then test in our experimental paradigm.

In the insusceptible condition, on the other hand, we hypothesize that behavior may be driven by a desire to punish the partner who violates the reciprocity norm by ignoring their opinion. We also consider another possible motivation for refusing to take influence from an insusceptible partner. Several studies in behavioral economic (ultimatum game in particular) have shown that one reason why people reject unfair offers is because they want to send the signal that they will not be easily dominated and thereby reject an inferior social status compared to their peers⁵⁻⁷. It is also possible that such confrontational stance helps the rejecting agent improve their self-efficacy. In summary, two different motives may drive people ignore the non-reciprocating partner: first, a nonreciprocal partner may be seen as violating the norm and people may wish to punish norm violators⁸. Second, people might perceive social influence as proxy for social status and refuse to accept an inferior, submissive status.

We have now updated the introduction (lines 83-88) and discussion (lines 251-285) to clarify these points.

Finally, we thank the reviewer for the positive assessment of our normative and informational distinction. We have elaborated this distinction in more detail by considering human social decision making under the informational assumption (lines 52-77) and we now also elaborate in the discussion why our results cannot be explained by this assumption (lines 240-250).

4- Does the asymmetry mentioned in comment 1 replicate in experiment 2a?

Since in experiment 2A, each participant had only one partner whose behavior changed from susceptible to insusceptible or vice versa, we did not ask our participants to rate how much they liked their partner as it was the same partner.

5- It seems that the influence the insusceptible partner can take is randomly selected between 0-0.3 but that the average taken in experiment 1 is 0.3. Is this a typo?

We thank the reviewer for drawing our attention to this point and we apologize for the misleading information in the original manuscript. The influence of our participants on the insusceptible partner was slightly different between experiments 1 and 2. We have now updated the method section (Constructing Partner) to clarify this point and to report the full distribution of the participants' influence on partners for experiment 1 and 2 separately (lines 404-408).

References:

1. Gouldner, A. W. The norm of reciprocity: A preliminary statement. *Am. Sociol. Rev.* 161–178 (1960).
2. Hertz, U. *et al.* Neural Computations Underpinning The Strategic Management Of Influence In Advice Giving. *bioRxiv* 121947 (2017).
3. Campbell-Meiklejohn, D. K., Bach, D. R., Roepstorff, A., Dolan, R. J. & Frith, C. D. How the opinion of others affects our valuation of objects. *Curr. Biol.* **20**, 1165–1170 (2010).
4. Izuma, K. & Adolphs, R. Social manipulation of preference in the human brain. *Neuron* **78**, 563–573 (2013).
5. Xiao, E. & Houser, D. Emotion expression in human punishment behavior. *Proc. Natl. Acad. Sci. U. S. A.* **102**, 7398–7401 (2005).
6. Yamagishi, T. *et al.* Rejection of unfair offers in the ultimatum game is no evidence of strong reciprocity. *Proc. Natl. Acad. Sci.* **109**, 20364–20368 (2012).

7. Burnham, T. C. High-testosterone men reject low ultimatum game offers. *Proc. R. Soc. Lond. B Biol. Sci.* **274**, 2327–2330 (2007).
8. Cooter, R. Do Good Laws Make Good Citizens? An Economic Analysis of Internalizing Legal Values. (2000).

REVIEWERS' COMMENTS:

Reviewer #1 (Remarks to the Author):

I think the revised version is much improved. The only thing I would like clarified is how the authors dealt with the standard errors - since there are several observations from each individual, are the standard errors clustered on the individual for all the analysis? "All standard errors were calculated across subjects" is still unclear to me.

Reviewer #2 (Remarks to the Author):

The authors have dealt with the comments that I raised.

Reviewer #3 (Remarks to the Author):

I do not have any additional comments

REVIEWERS' COMMENTS:

Reviewer #1 (Remarks to the Author):

I think the revised version is much improved. The only thing I would like clarified is how the authors dealt with the standard errors - since there are several observations from each individual, are the standard errors clustered on the individual for all the analysis? "All standard errors were calculated across subjects" is still unclear to me.

All error bars were computed across participants. In situations where we had multiple measurements from each participant, we first computed the mean value for each participant, and then computed the standard error across the participants' mean. We explain this now in the revised manuscript accordingly, i.e. in the corresponding figure legends of Figure 2 and 3 and in more detail in the Methods section in lines 427-430.

Reviewer #2 (Remarks to the Author):

The authors have dealt with the comments that I raised.

Reviewer #3 (Remarks to the Author):

I do not have any additional comments